# Sociodemographic and clinical characteristics of people with ostomy and the adaptive domains of Roy's theory: A cross-sectional study

**Suenia Silva de Mesquita Xavier**[1], **Lays Pinheiro de Medeiros**[1], **Alcides Viana de Lima Neto**[2], **Isabelle Pereira da Silva**[1], **Silvia Kalyma Paiva Lucena**[1], **Adriana Catarina de Souza Oliveira**[3], **Rhayssa de Oliveira Araújo**[1], **Isabelle Katherinne Fernandes Costa**[1]*

**1** Departamento de Enfermagem, Universidade Federal do Rio Grande do Norte, Natal, RN, Brazil,
**2** Faculdade de Ciências da Saúde, Santa Cruz, RN, Brazil, **3** Departamento de Enfermeria, Universidad Católica San Antón de Murcia, Murcia, Murcia, Spain

☯ These authors contributed equally to this work.
* isabelle.fernandes@ufrn.br

**Data Availability Statement:** All files are available from the OSF database (accession number DOI 10.17605/OSF.IO/3TKPH).

## Abstract

### Introduction

The adaptation of people with ostomies may be associated with and affected by sociodemographic and clinical factors. The present study aimed to investigate the association between the sociodemographic and clinical characteristics and the adaptation of people with an intestinal stoma.

### Method

An analytical study, carried out through an interview with 200 patients with ostomy for five months. For that, it was applied to scale for the level of adaptation of ostomy patients to measure the physiological domains, self-concept, role function and interdependence and a questionnaire was used in which sociodemographic and clinical information. Descriptive and multivariate analyses were performed to test the study hypothesis.

### Results

The study pointed out statistically significant associations with male sex, age group below 60, low education level, Stoma time less than one year, below one minimum wage, temporary permanence criteria and presence of complications relation to low scores of adaptation.

### Conclusions

The association of sociodemographic and clinical factors with the measured adaptive modes provides important information for the planning of nursing care and other care providers, since it directs actions to the aspects that give greater adaptive difficulty to people with stomas and which are the focus of care nursing to this clientele.

**Funding:** Funding by CNPq, process number 442895/2019-4 PCD 2019. The funders had no role in study design, data collection and analysis, decision to publish, or preparation of the manuscript.

**Competing interests:** The authors have declared that no competing interests exist.

## Introduction

A person with a ostomy is someone who has undergone surgery that results in the making of a stoma. Stoma means the mouth or opening of any hollow viscera to the outside of the body through a surgical act, in order to supply the need for elimination or feeding [1]. The main reasons for ostomy surgery are cancer or inflammatory bowel disease, incontinence, infections, constipation, or complications from surgery [2, 3].

The population with an ostomy has important epidemiology in several countries and requires prominence in public health. In a survey carried out in the United States of America, it was estimated that more than 750,000 people live with an ostomy and that 100,000 surgeries are performed annually to make new stomas [4].

Living with an ostomy, regardless of its cause, can cause physical and psychological impacts that affect patients, family and friends [5]. The change of the disposal site and the use of a collection bag attached to the abdomen for elimination represents the greatest physical difficulty faced, in addition to the possibility of leakage of effluents, evacuation of noise and intestinal gases through the bag during social events producing constraints [6]. Factors such as shorter stoma-making time, lack of religious practice, and lack of a partner negatively influenced the population's quality of life [7].

A study that analyzed the correlation between adaptation and resilience, under the theoretical framework of Roy's Adaptation Model, found that participants more adapted to the ostomy had higher levels of resilience than those with low adaptation. Roy's Model is a guideline for understanding the level of adaptation of the person with an ostomy, coping mechanisms, and environmental factors that interfere in this process [8].

The use of instruments to assess adaptation is essential in the health area, since the adaptive process of people with an ostomy is complex. Thus, adaptation can be understood, according to Callista Roy, as the person's adaptive responses to multidimensional stressors in a constant system that has nursing as the main facilitating agent of this process [9].

People with ostomies take variable time to achieve adaptation in different aspects of life depending on subjective factors and sociodemographic, clinical and health characteristics. Knowing these aspects can help health professionals to carry out the necessary follow-up and guidance, which are decisive in the adaptive process of people with stomas [10, 11]. Studies are needed to assess the adaptation and the characteristics of people with an intestinal ostomy, as well as the relationship between both dimensions to provide assistance to this population.

Therefore, this study aimed to investigate the association between the sociodemographic and clinical characteristics and the adaptation of people with an intestinal stoma.

## Materials and methods

### Ethical considerations

The ethical aspects related to research with human beings were observed, with authorization request, signing the free and informed consent form (IC). The study was authorized by the Research Ethics Committee of the Federal University of Rio Grande do Norte, under opinion number 1,527,460, and Presentation Certificate for Ethical Appreciation (CAAE) number 55191516.8.0000.5537.

Prior to the interviews, participants were provided with guidelines and general information regarding questions, risks and benefits before signing the Free and Informed Consent Form. Consent was obtained with a written signature; one copy being given to the participant and the other kept by the researcher. The precepts of ethics, good clinical practice and volunteer status were respected, according to the Declaration of Helsinki.

## Study design and location

This is an analytical study carried out at the Child and Adult Rehabilitation Center of Rio Grande do Norte (CRI / CRA) from July 17, 2017 to October 31, 2017. The Center is localized in Natal in the state of Rio Grande do Norte/ Brazil and is a reference for care for people with an ostomy through multidisciplinary care for health needs, as well as distribution of collection bags.

## Population and sample

For inclusion of the participants, the following criteria were adopted: having only an intestinal stoma rather than; be 18 years of age or older. The following patients were excluded from the study: those with previous diagnosis of major depression, neuropsychopathies or other serious mental illness registered in clinical records, and individuals who had other stoma types were excluded.

When considering a population (N) of 410 patients registered at the center in 2017, after performing the sample calculation, with confidence level of 95.0% and margin of error of 5.0% ($n = N.Z^2.p.(1-p) / Z^2.p.(1-p) + e^2.N-1$) a sample (n) of 199 individuals were required to obtain statistical significance. From Monday to Friday the researchers went to the center until they reached a sample of 200 people (consecutive chosen).

## Instruments and variables

For data collection, a questionnaire was used in which sociodemographic and clinical information was approached, such as: age, gender, marital status, scholarity, income, ostomy time, permanence criterion and complications.

The scale for the level of adaptation of ostomy patients (SLAOP) was also applied, an instrument to measure the physiological domains, self-concept, role function and interdependence, which contemplates the elements of Roy Adaptation Model, in order to assess the adaptation of the person with an ostomy [6, 7]. This scale includes 32 items, distributed in 4 domains, namely: physiological, self-concept, role function and interdependence. For each item, there is a score from 0 to 4 regarding the participant's responses, which can be: I totally agree; partially agree; indifferent; I partially disagree and I totally disagree. At the end, a score is assigned to classify the level of adaptation, the minimum score of the instrument's total score is zero and the maximum value of 128, see Table 1 [11, 12].

## Data collection

The research coordinators provided training for all interviewers regarding the application of the instruments selected for the study before performing the data collection. They were administered by undergraduate and graduate students of the Federal University of Rio Grande do Norte. The interview was conducted in a private room, conducted by the researcher in an interview format with questions guided by the instruments.

**Table 1. Presentation of scores in adaptive mode [11, 12].**

| Domains | Quantity of items | Score (minimum-maximum) |
|---|---|---|
| physiological | 7 | 0–28 |
| Self-concept | 17 | 0–68 |
| role function | 4 | 0–16 |
| interdependence | 4 | 0–16 |
| **Total** | 32 | 0–128 |

## Statistical analysis

Once the data was obtained, they were summarized using the techniques of descriptive and inferential statistics. The relationships between the domains of the scale for the level of adaptation of ostomy patients and the sociodemographic, clinical and health variables were investigated to verify the levels of adaptation of the respondents according to the results of the scale.

To evaluate this relationship, the Statistical Package for the Social Sciences (SPSS) version 20.0, statistical program and the Mann-Whitney test were used. The significance level of p-value $\leq$ 0.05 was adopted.

The study of the determinants of adaptation was carried out by means of linear regression analysis applied to Stepwise criteria, that verifies which of the independent variables are more determinable as determinants of the total score of adaptation. For this, because of the absence of normal data distribution, the scores were transformed into posts (posts). The independent variables used in all regression models were those that had a relationship between sociodemographic, clinical and health characteristics and the domains of the scale for the level of adaptation of ostomy patients, obtaining a p $\leq$ 0.10.

## Results

Regarding the main characteristics of the study participants, male people predominated (128; 64.0%), with a partner (116; 58.0%), incomplete primary education (50; 25.0%), retirees / beneficiaries (156; 78.0%), with income of up to one minimum wage (167; 83.5%), with the presence of colostomy (182; 91.0%) and with more than one year of ostomy (137; 68, 5%), see Table 2.

The level of adaptation was low. When associating the scores of the instrument and the sociodemographic variables, using the Mann Whitney test, statistically significant differences were found in relation to sex, age group, income and education, as described in Table 3.

It was found that females had better scores in all domains evaluated and in the general score of the instrument with statistical significance in the role function mode (p = 0.004), interdependence (p = 0.014) and general score (0.045).

Regarding the age range, those over 60 obtained the best scores in all the domains evaluated and in the general score, with statistical significance in the physiological mode (p = 0.035), role function (p <0.001) and general score (p = 0.035). Despite not showing significant associations with regard to marital status, participants who lived with a partner had better scores when compared with those without a partner.

As for income, those who received more than one minimum wage per month had higher scores in the assessed domains, with statistical significance in all modes and in the general score. Those who had a higher level of education, represented by secondary and higher education, obtained better adaptation scores, especially in the interdependence mode (p = 0.015).

When the clinical and health characteristics were associated with the instrument scores, there was statistical significance in the characteristics of ostomy time, permanence criteria, presence of complications with emphasis on allergy, presence of injury, leakage, itching and redness, as shown Table 4.

With regard to the time of ostomy, it was found that people with more than one year of ostomy had better scores in all domains and in the general score (p = 0.001), with statistical significance in the physiological domains (p = 0.017), self-concept (p = 0.001) and interdependence (p = 0.008).

Regarding the permanence criterion, it was found that people with permanent stomas had better scores in all domains and in the general score, with statistical significance in all. People with ostomies who presented complications had lower scores in all domains and in the general

**Table 2. Sociodemographic and clinical characterization of research participants.**

| Variables | | N | % |
|---|---|---|---|
| **Gender** | Male | 128 | 64,0 |
| | Female | 72 | 36,0 |
| **Age Range** | Up to 60 years | 121 | 60,5 |
| | > 60 years | 74 | 37,0 |
| | Didn't answer | 5 | 2,5 |
| **Marital status** | With partner | 116 | 58,0 |
| | Without patner | 84 | 42,0 |
| **Scholarity** | Illiterate | 20 | 10,0 |
| | Literate | 27 | 13,5 |
| | Incomplete elementary | 50 | 25,0 |
| | education | 26 | 13,0 |
| | Complete elementary education | 25 | 12,5 |
| | Incomplete high school | 37 | 18,5 |
| | Complete high school | 5 | 2,5 |
| | Incomplete higher education | 10 | 5,0 |
| **Occupation** | Retirees/beneficiaries | 156 | 78,0 |
| | Unemployed | 19 | 9,5 |
| | Housewife | 13 | 6,5 |
| | Working | 12 | 6,0 |
| **Income** | Up to 1 minimum wage | 167 | 83,5 |
| | > 1 minimum wage | 33 | 16,5 |
| **Type of ostomy** | Colostomy | 182 | 91,0 |
| | Ileostomy | 18 | 9,0 |
| **Ostomy time** | > 1 year | 137 | 68,5 |
| | Up to 1 year | 63 | 31,5 |
| **Permanence** | Temporary | 111 | 55,5 |
| | Definitive | 89 | 44,5 |
| | **Total** | **200** | **100,0** |

score with statistical significance in all of them. The following stood out: leakage (62.5%), redness (58%), itching (41%), allergies (37%) and injuries (34.5%).

The variables education, sex, income, age, time of ostomy, length of stay and complications were subjected to the multiple linear regression analysis model, which indicated a significant correlation, with $p \leq 0.01$ and R = 0.201. This means that all these variables obtained 20.2% of the variance related to the adaptation scores.

## Discussion

The sociodemographic association demonstrated that males had worse scores in the role function and interdependence modes. This refers to the differences in the process of coping with the ostomy.

A study carried out to compare the psychosocial adjustment between men and women, found that the scores obtained by females (43.45 ± 12.81) were higher than those of males (37.68 ± 12.96), in which men were most affected in the domains related to ostomy acceptance, social engagement and negative feelings, because of lack of family support and dependence on stoma care [13]. Women have a greater degree of independence from the ostomy, which can favor psychosocial aspects and the adaptation process. The importance of social support is highlighted to strengthen the autonomy and adaptation of these people, which can be provided by family, friends, partners and professionals [6].

**Table 3. Result of the general comparison of the instrument's domains with the sociodemographic variables using the Mann Whitney test.**

| Variables | Domains* | | | | | | | | | |
|---|---|---|---|---|---|---|---|---|---|---|
| | 1 | | 2 | | 3 | | 4 | | Total | |
| | mean (SD) | p-value | mean (SD) | p-value | mean (SD) | p-value | mean (SD) | p-value | mean (SD) | p-value |
| **Gender** | | | | | | | | | | |
| Female | 10,9 (7,1) | 0,619 | 41,1 (17,7) | 0,053 | 7,2 (5,4) | 0,004 | 8,0 (5,0) | 0,014 | 67,3 (30,9) | 0,045 |
| Male | 10,2 (6,8) | | 36,5 (16,3) | | **5,1 (4,9)** | | **6,1 (5,1)** | | **58,1 (29,5)** | |
| **Age Range** | | | | | | | | | | |
| Up to 60 years | **9,5 (6,1)** | 0,035 | 37,1 (16,1) | 0,104 | **4,9 (4,6)** | <0,001 | 6,6 (5,2) | 0,199 | **58,2 (28,0)** | 0,035 |
| > 60 years | 12,9 (7,9) | | 40,6 (18,3) | | 7,8 (5,6) | | 7,4 (4,9) | | 68,0 (33,1) | |
| **Marital status** | | | | | | | | | | |
| With partner | 10,9 (6,7) | 0,252 | 38,6 (16,6) | 0,697 | 5,9 (5,3) | 0,935 | 7,2 (5,1) | 0,102 | 62,8 (29,4) | 0,439 |
| Without patner | 9,9 (7,2) | | 37,6 (17,4) | | 5,9 (5,2) | | 6,1 (5,2) | | 59,6 (31,5) | |
| **Income** | | | | | | | | | | |
| up to 1 minimum wage | **9,75 (6,3)** | 0,007 | **36,90 (16,7)** | 0,018 | **5,14 (4,7)** | <0,001 | **6,46 (5,2)** | 0,046 | **58,26 (28,7)** | 0,002 |
| > 1 minimum wage | 14,0(8,7) | | 44,2 (17,2) | | 9,8 (5,8) | | 8,4 (4,6) | | 76,5 (33,5) | |
| **scholarity** | | | | | | | | | | |
| Elementary School | 10,0 (6,9) | 0,243 | **36,7 (17,0)** | 0,097 | 5,4 (4,9) | 0,154 | **6,0 (5,0)** | 0,015 | **58,2 (29,7)** | 0,06 |
| High school and higher | 11,2 (7,0) | | 40,6 (16,6) | | 6,7 (5,7) | | 7,9 (5,1) | | 66,4 (30,7) | |

*1. Physiological; 2. Self-concept; 3. Paper function; 4. Interdependence.

Another important aspect to be considered, with regard to the differences between men and women is sexuality, which represents something inherent to human experience and has an important impact on both sexes, after the ostomy. The presence of the stoma negatively influences self-image, since there is a rejection associated with body image due to the presence of the collecting device [14].

Roy's model exposes the roles that people play in society together with the social relationships associated with others, influencing the ability of individuals to heal or maintain health. Thus, the difficulties associated with the functions performed may constitute stimuli that cause an ineffective adaptation [15].

In this study, despite not showing statistical significance, people who lived with partners had better averages. The presence of a partner can represent important support in facing the

**Table 4. Result of the general comparison of the instrument's domains with the clinical and health variables using the Mann Whitney test.**

| Domains* | Measures | Variables | | | | | |
|---|---|---|---|---|---|---|---|
| | | Ostomy Time | | Permanence Criterion | | Complications | |
| | | Up to 1 year | More than 1 year | Definitive | Temporary | yes | No |
| 1 | **Mean (SD)** | **8,7 (6,3)** | 11,3 (7,1) | 11,7 (6,7) | **9,5 (6,9)** | **9,5 (6,4)** | 16,0 (7,0) |
| | ***p*-value** | 0,017 | | 0,015 | | <0,001 | |
| 2 | **Mean (SD)** | **32,2 (17,5)** | 41,0 (15,9) | 43,0 (17,0) | **34,4 (15,9)** | **37,2 (16,9** | 44,0 (15,7) |
| | ***p*-value** | 0,001 | | <0,001 | | 0,046 | |
| 3 | **Mean (SD)** | 4,9 (4,6) | 6,4 (5,4) | 7,3 (5,3) | **4,9 (4,9)** | **5,5 (5,1)** | 8,5 (5,4) |
| | ***p*-value** | 0,071 | | 0,001 | | 0,004 | |
| 4 | **Mean (SD)** | **5,4 (4,7)** | 7,4 (5,2) | 7,9 (4,6) | **5,9 (5,3)** | **6,4 (5,0)** | 9,1 (5,1) |
| | ***p*-value** | 0,008 | | 0,004 | | 0,005 | |
| Total | **Mean (SD)** | **51,2 (44,0)** | 66,1 (29,4) | 69,9 (29,6) | **54,6 (29,1)** | **58,5 (29,8)** | 77,6 (28,2) |
| | ***p*-value** | 0,001 | | <0,001 | | 0,001 | |

challenges of living with a stoma. Furthermore, the stoma can strengthen coexistence and communication between partners. In this context, sexual aspects are substantially modified by the presence of the ostomy, but can be overcome by strategies such as changing positions during sexual activities, emptying and hiding the collection bag and support from health professionals [16].

In this sense, it is important that the nurse is attentive to all these aspects and performs an appropriate approach for a person with an ostomy, specifying the particularities of each individual and in the aspects that differ or not from the stoma mode.

Regarding an age group, elderly people have better results in physiological mode and role function. In contrast, studies that show the elderly have shown greater vulnerability in the aging process, developing more difficulties in relation to adaptation to the stoma [17, 18].

Aging associated with changes in the new health condition does not impose a state of incapacity or dependence in the face of self-care actions, for the elderly. Thus, it is important to give due consideration to the social roles exercised, as well as performing self-care actions, allowing them to provide a state of well-being and autonomy [19].

Adaptation in Role Function is one of the difficulties imposed by a new life after stoma construction. People with an ostomy fear embarrassment and present difficulties, in body image, mainly if they are young. The insecurity that this procedure causes in some is common, the fear of leaks, flatulence and causing discomfort in the people around them was reported in another study [20, 21].

Regarding marital status, it was observed that the participants who had a partner obtained better scores than those without partners. This highlights the importance of the support of a partner, who supports the acceptance and overcoming of problems arising from the stoma. The difficulties with an ostomy are shared by both in the relationship, and the companion of people with an ostomy must be integrated into the nursing care provided, as an essential member of the social support network [6].

As for income, those who received more than one minimum wage per month had higher scores, with statistical significance in all modes and in the general score. These results reflect the influence of a source of income on the well-being of people with an ostomy, making it possible to contribute to family expenses, guaranteeing autonomy and subsidies for the purchase of inputs and leisure programs [22].

It can be seen in this study and in others, that most people have low income, as well as those who do not work or have pensions or benefits [23]. There are difficulties in returning to remunerated activities, which could improve financial acquisition, but that is a challenge for the person with an ostomy [24].

In Brazil, there are initiatives to encourage this reintegration of people with an ostomy into the labor market, with emphasis on Law No. 8,213, of July 24, 1991, which aims to integrate people with disabilities, as well as create mandatory vacancies in companies designed for this purpose. However, many of the population are unaware of these opportunities [25].

As for the level of education, it was found that the participants who had a higher level of education obtained better scores, with a significant association with the interdependence mode. Studies show that most people with ostomy have low education, as in this research [22, 23]. It is understood that better levels of education provide broader knowledge about their rights, as well as health care, in addition to clarification about themselves, which may favor autonomy and security.

Regarding the duration of stoma, it was found that those who had more than one year showed a significant association with the physiological, self-concept and interdependence modes. A study conducted to assess self-esteem found that people who had less than four years of ostomy had lower levels of self-esteem and body image [26]. Accordingly, another study

showed that people with up to two years of stoma had greater limitations and psychosocial impacts than those persisting for three years or more [27].

The adaptation process is something gradual, which is built through the social support of the family, partner and health professionals, in addition to the acquired experiences, in which people with a stoma gradually learn about how to live with the ostomy. This helps to be able to develop strategies to overcome the difficulties arising from the process [28].

As for the permanence criterion, it was evidenced that people with permanent ostomies had better scores in all domains. In line with these results, another study that evaluated self-efficacy in people with temporary ostomies found that most (85.6%) had low to moderate self-efficacy in the total mean scores (78.55 ± 14.72) [29].

This can occur due to the negative impacts caused by the surgery and the insecurity regarding the reversal procedure, which generates greater anxiety and less security to perform self-care actions. In this sense, social support from family members and health professionals is important to encourage the learning and adaptation of these people. Coexistence with other people with permanent stomas, who have been in this condition for a longer time, can help in sharing information and adaptation mechanisms [29].

Regarding the type of scholarship used, there was no significant association. The difficulties in continuing with the bag that best adapts to the type of stoma are highlighted, since many start using one type of bag, they need to interrupt and make use of another that is available.

Brazilian law ensures the availability of free collection equipment in an appropriate quantity, as well as other necessary inputs, but in practice there is a lack of device [24]. In addition to this, difficulties regarding the quality of the device, which were frequently mentioned by the research participants.

The results also demonstrated that the people who developed complications with the stoma, obtained lower scores in all domains and in general, with statistical significance in all of them. In addition, the most significant complication was leakage of fecal content.

The presence of complications has multiple impacts on the quality of life of the person with an ostomy, with damages that affect the physical aspect, which require specific care and costly expenses. Psychoemotional aspects are also affected, with an increase in negative feelings towards the stoma [30].

A study carried out to assess the effect of complications on the quality of life of people with an ostomy, concluded that people with irritant contact dermatitis were the most affected and the statistics were lower when compared to those without complications. Of those who developed complications, many did not receive care or support with these problems [31].

Many complications, such as contact dermatitis, can be avoided through professional monitoring from the preoperative period, with effective self-care guidelines, such as clipping, correct placement of the bag, use of skin protection materials and use of peristomal devices compatible with the type of stoma [32].

Regarding the studied linear regression, such as school variables, sex, income, age group, duration, duration of stay and complications obtained, correlations obtained, with a prediction of 20.2% of the variation in adaptation. Note that some sociodemographic characteristics, such as sex and age group, are unchangeable and must be considered when planning care.

A study carried out in California found a relationship between younger age of people with stomas and the presence of skin complications, leaks from the collection bag and a greater number of reports about difficulties in self-care. Factors such as lower income can also affect the challenges experienced in the adaptive process [22].

Some interventions can be considered by the nurse to work on the management of the ostomy, considering the predictors that influence the adaptation process. For people with low income and financial difficulties to get around, alternative means of communication and

health education with trained professionals, such as telehealth, can be considered through monitoring via videoconferences and telephones, especially in the recent period after surgery, considering the influence ostomy time [33–35].

Educational interventions with practical training sessions and encouraging autonomy for people with stomas are other ways to manage adaptive problems. In this program, it is important to consider the educational level of the people and the characteristics of the participants, in order to plan the best approach that can be centered on solving problems and exchanging experiences [33]. Simulation devices that assist in demonstrations and in the learning of the population [36].

Therefore, there are significant associations between sociodemographic and clinical characteristics and adaptation. Educational nursing interventions that consider these aspects are necessary to assist the adaptive process of people with ostomy.

## Study limitations

The study was carried out in a single center, which limits the generalization of findings in other health contexts. Furthermore, due to the study design, it is not possible to establish causalities, which could further explain the relationships between the variables.

## Conclusion

The study pointed out statistically significant associations in relation to low scores of adaptation between the male sex and the role function and interdependence modes; age group below 60, physiological mode and role function; low education level and interdependence domain. Stoma time less than one year also had significant associations with low adaptive scores in the physiological, self-concept and interdependence domains. Regarding income below one minimum wage, temporary permanence criteria and presence of complications, all obtained low scores associations across all adaptive domains.

The association of sociodemographic and clinical characteristics with the adaptive domains of the Roy Adaptation Model, provides important information for the planning of nursing care, directs educational actions to the aspects that give greater adaptive difficulty to people with stomas and that constitute the focus of nursing care for this clientele.

This study contributes to the advancement of the theoretical and scientific knowledge of nursing as it investigates factors that are associated with the adaptive process of the population with an ostomy.

## Author Contributions

**Conceptualization:** Lays Pinheiro de Medeiros, Isabelle Katherinne Fernandes Costa.

**Data curation:** Suenia Silva de Mesquita Xavier, Lays Pinheiro de Medeiros, Isabelle Katherinne Fernandes Costa.

**Formal analysis:** Suenia Silva de Mesquita Xavier, Lays Pinheiro de Medeiros, Isabelle Katherinne Fernandes Costa.

**Funding acquisition:** Isabelle Katherinne Fernandes Costa.

**Investigation:** Lays Pinheiro de Medeiros, Isabelle Pereira da Silva, Silvia Kalyma Paiva Lucena, Isabelle Katherinne Fernandes Costa.

**Methodology:** Suenia Silva de Mesquita Xavier, Isabelle Katherinne Fernandes Costa.

**Project administration:** Lays Pinheiro de Medeiros, Isabelle Katherinne Fernandes Costa.

**Supervision:** Alcides Viana de Lima Neto, Adriana Catarina de Souza Oliveira, Rhayssa de Oliveira Araújo, Isabelle Katherinne Fernandes Costa.

**Writing – original draft:** Suenia Silva de Mesquita Xavier, Lays Pinheiro de Medeiros, Alcides Viana de Lima Neto, Isabelle Pereira da Silva, Silvia Kalyma Paiva Lucena, Adriana Catarina de Souza Oliveira, Rhayssa de Oliveira Araújo, Isabelle Katherinne Fernandes Costa.

**Writing – review & editing:** Adriana Catarina de Souza Oliveira, Rhayssa de Oliveira Araújo, Isabelle Katherinne Fernandes Costa.

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
