## [Decision Letter · Decision Letter 0]

22 Feb 2024

PONE-D-23-42687SOCIODEMOGRAPHIC AND CLINICAL CHARACTERISTICS OF PEOPLE WITH OSTOMY AND THE ADAPTIVE DOMAINS OF ROY'S THEORY: A cross-sectional studyPLOS ONE

Dear Dr. Costa, Thank you for submitting your manuscript to PLOS ONE. After careful consideration, we feel that it has merit but does not fully meet PLOS ONE’s publication criteria as it currently stands. Therefore, we invite you to submit a revised version of the manuscript that addresses the points raised during the review process.

We look forward to receiving your revised manuscript.

Kind regards,

Marco Clementi, Assistant Professor

Academic Editor

PLOS ONE

Journal Requirements:

"Funding by CNPq, process number 442895/2019-4 PCD 2019"

3. Please expand the acronym “CNPq” (as indicated in your financial disclosure) so that it states the name of your funders in full.

Additional Editor Comments:

The paper requires major revision to be re-evaluated

Reviewers' comments:

Reviewer's Responses to Questions

**Comments to the Author**

1. Is the manuscript technically sound, and do the data support the conclusions?

Reviewer #1: Yes

Reviewer #2: Partly

2. Has the statistical analysis been performed appropriately and rigorously? 

Reviewer #1: Yes

Reviewer #2: I Don't Know

3. Have the authors made all data underlying the findings in their manuscript fully available?

Reviewer #1: Yes

Reviewer #2: No

4. Is the manuscript presented in an intelligible fashion and written in standard English?

Reviewer #1: Yes

Reviewer #2: Yes

5. Review Comments to the Author

Reviewer #1: The association of sociodemographic and clinical factors with the measured adaptive modes provides important information for the planning of nursing care and other care providers, since it directs actions to the aspects that give greater adaptive difficulty to people with stomas and which are the focus of care nursing to this clientele.

The manuscript demonstrates creativity and novelty. However, there are several suggestions for improvement:

1. Please specify the version of SPSS used for statistical analysis.

2. How was the sample size for the survey calculated? Increasing the sample size can strengthen the evidence supporting the conclusions.

Reviewer #2: The authors are to be congratulated on performing this study in a neglected area of healthcare. Although much is written about surgical aspects of stoma creation and its physical complications, much less is known about the social and personal effects of living with a stoma. The paper reports a single centre series of patients with stomas who were interviewed regarding their level of adaption to life with a stoma across a range of sociodemographic domains.

In general, the paper is quite easy to understand. However, there are multiple grammatical and typographical errors which should be corrected. A list is given at the end of this peer review report and should all be addressed prior to publication if the paper is accepted by the editor.

Issues for consideration-

1 – This is a single centre study – a tertiary care centre. It is therefore difficult to know how applicable these results would be to patients in different health settings.

2 – The authors do not explain how they chose the 200 participants for the study. They state that 410 patients were registered with the centre at this time period and then mention a sample of 199 eligible patients. I am not sure if they mean that 199 patients were required to obtain statistical significance, or whether there were only 199/410 patients who were eligible. Furthermore, it is not clear if selected patients were consecutive, or randomly chosen, or selectively chosen. This needs to be clarified, so that the results can be interpreted in the light of the included patients.

3 – In the section on “instruments and variables” – last paragraph. The final two sentences do not make sense. The penultimate sentence on the scoring system is very confusing and makes no sense. This needs to be either rewritten completely or a table with the scoring system should be inserted.

4- Results section – this needs to be improved with the addition of a new Table 1 which gives the demographic data on the included patients. This would be the baseline against which the following results could be analysed. It is essential to set out how many male/female patients, ages, etc. If a new Table 1 is inserted, then the numbers of the other tables will all be moved on and this should be altered in the text as well as on the legends.

5- In the discussion the authors state that lack of family support links with poor scores across several domains. However, the results are only for lack of a partner, rather than all family support. This is a subtle difference as many family members/friends can provide support whether the patient has a partner or not.

6- the sentence on Study Limitations does not make sense. There are many limitations to this study and at least one paragraph should be devoted to describing them in more detail.

TYPOGRAPHICAL AND GRAMMATICAL ERRORS

Page 1 TITLE – There are two : after “Theory” – please delete one of them.

ABSTRACT -In the sentence regarding Results- Please change from “significant associations in the male sex” to “significant associations with male sex”

INTRODUCTION

Page 2 – paragraph 2 – there is a line ____________ before the first word in this paragraph. Please delete.

Page 2 - Paragraph 3 – line 3 – change “for eliminations” to “for elimination”

Page 2 – Paragraph 3 – line 5 – change “elimination” to “evacuation”

Page 2 – Paragraph 3 – line 5 – remove the commas after “bag” and after “social events”

Page 2 – Paragraph 4 – Line 1 – should start with “A study” not just “Study”

Page 2 – Paragraph 5 – line 3 – Pleas delete “for” before “health professionals”

Page 2 – Paragraph 5 – line 5 – please change “this characteristics” to “the characteristics”

Page 2 – Paragraph 5 – line 7 – please change “support assistance” to “provide assistance”

MATERIALS AND METHODS

Page 3 – The title of the section should be “Materials and Methods” not “Material and Methods”

Ethical Considerations – second paragraph – please change “clarifications” to “information”

Ethical Considerations – second paragraph – line 2 – please change “the written signature of the term” to “a written signature”

Population and Sample section – line 2 please put a : rather than ; after “stoma”

Statistical analysis section – last paragraph – line 2 – change “are independent” to “of the independent”

Statistical analysis section – last paragraph – line 4 - change "by the absence” to “because of the absence”

RESULTS

Tabel 1 – there is a spelling mistake in the left column – It should be “Age Range” not “Age ranger”

In the results section – the penultimate paragraph – should be changed from-

“all of them.. they following stood out” to-

“all of them. The following stood out”

DISUCSSION

Paragraph 1 - Line 1 – change “male” to “males”

Paragraph 2 – Line 4 – change “because lack of” to “because of lack of “

Paragraph 6 – Line 1 – please change “see if elderly people have better results” to “elderly people had better results”

Paragraph 7- Line 2 – please delete Wince”

Paragraph 7 – Line 3 – please change “give importance” to “give due consideration”

Paragraph 8 – Line 2 – please change “the stoma construction. Colostomy people fear” to “stoma construction. People with an ostomy fear”

Paragraph 8 – Line 3 – please change “you’re” to “they are”

Paragraph 8 – Line 4 – please change “he fear of leaks” to “the fear of leaks”

Paragraph 9 – Line 3 – please change “favors” to “supports”

Paragraph 9 – Line 4 - please change “with ostomy” to “with an ostomy”

Paragraph 11 – Line 2 – please change “ as they do not work and have pensions” to “as those who do not work or have pensions”

Paragraph 11 – Line 3 – please change “but that become” to “but that is”

Paragraph 12 – Line 4 - please change “population, however many of them are unaware” to “However, many of the population are unaware of these opportunities”

Paragraph 14 – Last line – please change “those aged three years or older” to “those persisting for three years or more”

Paragraph 18 - line 1 – please change “scholarship” to “device”

Paragraph 19 – Line 2 – please change "inputs” to “devices” and change “scholarship” to “device

Paragraph 20 – Line 1 – please change “presented” to “developed”

Paragraph 20 – line 2 - please change “all of them, in addition” to “all of them. In addition”

Paragraph 22 – line 2 - please change “concluded, as people” to “concluded that people”

Paragraph 22- line 4 – please change “presented” to “developed”

Paragraph 22 – line 4-5 - please change “many were not attended to or received guidance for the care of injuries” to “many did not receive care or support with these problems”

Paragraph 24 – Line 4 – please change “not acceptable, but” to “unchangeable and”

Paragraph 24 – the last sentence does not make sense. It should either be explain clearly or removed.

Paragraph 25 – Line 1 – please delete “the” before “younger”

Paragraph 26 – the first sentence does not make sense - do the authors mean that the length of time that a stoma has been present does not affect self-care?

Last paragraph of results section – please change – “between characteristics sociodemographic and clinical and the adaptation” to “between sociodemographic and clinical characteristics and adaptation”

CONCLUSION-

Paragraph 1 – Line 4- please change “one year had” to “one year also had”

Paragraph 1 – Line 4 - please change "associations to” to “associations with”

Paragraph 1 – last line - please change “with all” to “across all”

Paragraph 2 – line 2 – please change “subsidies” to “information”

Paragraph 3 – Line 1 - please change “The study” to “This study”

6. PLOS authors have the option to publish the peer review history of their article (what does this mean?). If published, this will include your full peer review and any attached files.

Reviewer #1: No

Reviewer #2: No

---

## [Author Response · Author response to Decision Letter 0]

26 Feb 2024

Thank you very much for agreeing to consider our manuscript entitled “SOCIODEMOGRAPHIC AND CLINICAL CHARACTERISTICS OF PEOPLE WITH OSTOMY AND THE ADAPTIVE DOMAINS OF ROY'S THEORY: A cross-sectional study"

We are pleased to submit the revised version of the manuscript mentioned above. We would like to express our gratitude to the reviewers and the editor for their valuable feedback, which has significantly improved the quality of the article. In ANNEX 1, we have provided detailed responses to each comment made by the Reviewers and editor. All of the proposed changes have been incorporated into the revised text and highlighted in yellow.

We would be very grateful if you could consider our manuscript to be published in your journal. 

Thank you for your consideration of this manuscript.

Yours sincerely,

The authors

---

## [Decision Letter · Decision Letter 1]

6 Mar 2024

PONE-D-23-42687R1SOCIODEMOGRAPHIC AND CLINICAL CHARACTERISTICS OF PEOPLE WITH OSTOMY AND THE ADAPTIVE DOMAINS OF ROY'S THEORY: A cross-sectional studyPLOS ONE

Dear Dr. Costa,

Thank you for submitting your manuscript to PLOS ONE. After careful consideration, we feel that it has merit but does not fully meet PLOS ONE’s publication criteria as it currently stands. Therefore, we invite you to submit a revised version of the manuscript that addresses the points raised during the review process.

We look forward to receiving your revised manuscript.

Kind regards,

Marco Clementi, Assistant Professor

Academic Editor

PLOS ONE

**Comments to the Author**

1. If the authors have adequately addressed your comments raised in a previous round of review and you feel that this manuscript is now acceptable for publication, you may indicate that here to bypass the “Comments to the Author” section, enter your conflict of interest statement in the “Confidential to Editor” section, and submit your "Accept" recommendation.

Reviewer #1: All comments have been addressed

Reviewer #2: All comments have been addressed

2. Is the manuscript technically sound, and do the data support the conclusions?

Reviewer #1: Yes

Reviewer #2: Partly

3. Has the statistical analysis been performed appropriately and rigorously? 

Reviewer #1: Yes

Reviewer #2: No

4. Have the authors made all data underlying the findings in their manuscript fully available?

Reviewer #1: Yes

Reviewer #2: Yes

5. Is the manuscript presented in an intelligible fashion and written in standard English?

Reviewer #1: Yes

Reviewer #2: Yes

6. Review Comments to the Author

Reviewer #1: (No Response)

Reviewer #2: The authors are to be congratulated on performing this study in a neglected area of healthcare. Although much is written about surgical aspects of stoma creation and its physical complications, much less is known about the social and personal effects of living with a stoma. The paper reports a single centre series of patients with stomas who were interviewed regarding their level of adaption to life with a stoma across a range of sociodemographic domains.

In general, the paper is quite easy to understand. However, there are multiple grammatical and typographical errors which should be corrected. A list is given at the end of this peer review report and should all be addressed prior to publication if the paper is accepted by the editor.

Issues for consideration-

1 – This is a single centre study – a tertiary care centre. It is therefore difficult to know how applicable these results would be to patients in different health settings.

2 – The authors do not explain how they chose the 200 participants for the study. They state that 410 patients were registered with the centre at this time period and then mention a sample of 199 eligible patients. I am not sure if they mean that 199 patients were required to obtain statistical significance, or whether there were only 199/410 patients who were eligible. Furthermore, it is not clear if selected patients were consecutive, or randomly chosen, or selectively chosen. This needs to be clarified, so that the results can be interpreted in the light of the included patients.

3 – In the section on “instruments and variables” – last paragraph. The final two sentences do not make sense. The penultimate sentence on the scoring system is very confusing and makes no sense. This needs to be either rewritten completely or a table with the scoring system should be inserted.

4- Results section – this needs to be improved with the addition of a new Table 1 which gives the demographic data on the included patients. This would be the baseline against which the following results could be analysed. It is essential to set out how many male/female patients, ages, etc. If a new Table 1 is inserted, then the numbers of the other tables will all be moved on and this should be altered in the text as well as on the legends.

5- In the discussion the authors state that lack of family support links with poor scores across several domains. However, the results are only for lack of a partner, rather than all family support. This is a subtle difference as many family members/friends can provide support whether the patient has a partner or not.

6- the sentence on Study Limitations does not make sense. There are many limitations to this study and at least one paragraph should be devoted to describing them in more detail.

TYPOGRAPHICAL AND GRAMMATICAL ERRORS

Page 1 TITLE – There are two : after “Theory” – please delete one of them.

ABSTRACT -In the sentence regarding Results- Please change from “significant associations in the male sex” to “significant associations with male sex”

INTRODUCTION

Page 2 – paragraph 2 – there is a line ____________ before the first word in this paragraph. Please delete.

Page 2 - Paragraph 3 – line 3 – change “for eliminations” to “for elimination”

Page 2 – Paragraph 3 – line 5 – change “elimination” to “evacuation”

Page 2 – Paragraph 3 – line 5 – remove the commas after “bag” and after “social events”

Page 2 – Paragraph 4 – Line 1 – should start with “A study” not just “Study”

Page 2 – Paragraph 5 – line 3 – Pleas delete “for” before “health professionals”

Page 2 – Paragraph 5 – line 5 – please change “this characteristics” to “the characteristics”

Page 2 – Paragraph 5 – line 7 – please change “support assistance” to “provide assistance”

MATERIALS AND METHODS

Page 3 – The title of the section should be “Materials and Methods” not “Material and Methods”

Ethical Considerations – second paragraph – please change “clarifications” to “information”

Ethical Considerations – second paragraph – line 2 – please change “the written signature of the term” to “a written signature”

Population and Sample section – line 2 please put a : rather than ; after “stoma”

Statistical analysis section – last paragraph – line 2 – change “are independent” to “of the independent”

Statistical analysis section – last paragraph – line 4 - change "by the absence” to “because of the absence”

RESULTS

Tabel 1 – there is a spelling mistake in the left column – It should be “Age Range” not “Age ranger”

In the results section – the penultimate paragraph – should be changed from-

“all of them.. they following stood out” to-

“all of them. The following stood out”

DISUCSSION

Paragraph 1 - Line 1 – change “male” to “males”

Paragraph 2 – Line 4 – change “because lack of” to “because of lack of “

Paragraph 6 – Line 1 – please change “see if elderly people have better results” to “elderly people had better results”

Paragraph 7- Line 2 – please delete Wince”

Paragraph 7 – Line 3 – please change “give importance” to “give due consideration”

Paragraph 8 – Line 2 – please change “the stoma construction. Colostomy people fear” to “stoma construction. People with an ostomy fear”

Paragraph 8 – Line 3 – please change “you’re” to “they are”

Paragraph 8 – Line 4 – please change “he fear of leaks” to “the fear of leaks”

Paragraph 9 – Line 3 – please change “favors” to “supports”

Paragraph 9 – Line 4 - please change “with ostomy” to “with an ostomy”

Paragraph 11 – Line 2 – please change “ as they do not work and have pensions” to “as those who do not work or have pensions”

Paragraph 11 – Line 3 – please change “but that become” to “but that is”

Paragraph 12 – Line 4 - please change “population, however many of them are unaware” to “However, many of the population are unaware of these opportunities”

Paragraph 14 – Last line – please change “those aged three years or older” to “those persisting for three years or more”

Paragraph 18 - line 1 – please change “scholarship” to “device”

Paragraph 19 – Line 2 – please change "inputs” to “devices” and change “scholarship” to “device

Paragraph 20 – Line 1 – please change “presented” to “developed”

Paragraph 20 – line 2 - please change “all of them, in addition” to “all of them. In addition”

Paragraph 22 – line 2 - please change “concluded, as people” to “concluded that people”

Paragraph 22- line 4 – please change “presented” to “developed”

Paragraph 22 – line 4-5 - please change “many were not attended to or received guidance for the care of injuries” to “many did not receive care or support with these problems”

Paragraph 24 – Line 4 – please change “not acceptable, but” to “unchangeable and”

Paragraph 24 – the last sentence does not make sense. It should either be explain clearly or removed.

Paragraph 25 – Line 1 – please delete “the” before “younger”

Pragraph 26 – the first sentence does not make sense - do the authors mean that the length of time that a stoma has been present does not affect self-care?

Last paragraph of results section – please change – “between characteristics sociodemographic and clinical and the adaptation” to “between sociodemographic and clinical characteristics and adaptation”

CONCLUSION-

Paragraph 1 – Line 4- please change “one year had” to “one year also had”

Paragraph 1 – Line 4 - please change "associations to” to “associations with”

Paragraph 1 – last line - please change “with all” to “across all”

Paragraph 2 – line 2 – please change “subsidies” to “information”

Paragraph 3 – Line 1 - please change “The study” to “This study”

7. PLOS authors have the option to publish the peer review history of their article (what does this mean?). If published, this will include your full peer review and any attached files.

Reviewer #1: No

Reviewer #2: No

---

## [Author Response · Author response to Decision Letter 1]

11 Mar 2024

We are pleased to submit the revised version of the manuscript and we would like to express our gratitude to the reviewers and the editor for their valuable feedback, which has significantly improved the quality of the article. Reviewer 2's comments are the same as the previous review, and were already solved in the previous version. In ANNEX 1, we have provided detailed responses to each comment made by the Reviewers. All of the proposed changes have been incorporated into the revised text and highlighted in yellow.

---

## [Editor Report · Decision Letter 2]

28 Mar 2024

SOCIODEMOGRAPHIC AND CLINICAL CHARACTERISTICS OF PEOPLE WITH OSTOMY AND THE ADAPTIVE DOMAINS OF ROY'S THEORY: A cross-sectional study

PONE-D-23-42687R2

Dear Dr. Isabelle Katherinne Fernandes Costa,

We’re pleased to inform you that your manuscript has been judged scientifically suitable for publication and will be formally accepted for publication once it meets all outstanding technical requirements.

Kind regards,

Marco Clementi, Assistant Professor

Academic Editor

PLOS ONE

Additional Editor Comments:

All the comments were solved. No further changes are required. Thanks to the authors for their hard work.

---

## [Editor Report · Acceptance letter]

3 Apr 2024

PONE-D-23-42687R2 

PLOS ONE

Dear Dr. Costa, 

I'm pleased to inform you that your manuscript has been deemed suitable for publication in PLOS ONE. Congratulations! Your manuscript is now being handed over to our production team.

Kind regards, 

on behalf of

Dr. Marco Clementi 

Academic Editor

PLOS ONE